# Two-dimensional MXene membranes with biomimetic sub-nanochannels for enhanced cation sieving

Rongming Xu[1,2], Yuan Kang[3], Weiming Zhang [1,2] ✉, Bingcai Pan [1,2] ✉ & Xiwang Zhang [4] ✉

Membranes with high ion permeability and selectivity are of considerable interest for sustainable water treatment, resource extraction and energy storage. Herein, inspired by $K^+$ channel of streptomyces A (KcsA $K^+$), we have constructed cation sieving membranes using MXene nanosheets and Ethylenediaminetetraacetic acid (EDTA) molecules as building blocks. Numerous negatively charged oxygen atoms of EDTA molecules and 6.0 Å two-dimensional (2D) sub-nanochannel of MXene nanosheets enable biomimetic channel size, chemical groups and tunable charge density for the resulting membranes. The membranes show the capability to recognize monovalent/divalent cations, achieving excellent $K^+/Mg^{2+}$ selectivity of 121.2 using mixed salt solution as the feed, which outperforms other reported membranes under similar testing conditions and transcends the current upper limit. Characterization and simulations indicate that the cation recognition effect of EDTA and partial dehydration effects play critical roles in cations selective sieving and increasing the local charge density within the sub-nanochannel significantly improves cation selectivity. Our findings provide a theoretical basis for ions transport in sub-nanochannels and an alternative strategy for design ions separation membranes.

Biological ion channels are protein-based pores capable of regulating ion transport in living cells in response to external stimuli, leading to the ultra-selective transmembrane move of specific ions (e.g., $K^+$ channel with $K^+/Na^+$ above 10,000)[1,2]. To synthesize artificial ion channel membranes resembling biological ion channels not only facilitates the understanding of complex ion transport in bioprocesses but also readily enables critical industrial applications, including water treatment[3–5], resource extraction[6,7], energy conversion[8,9], and biosensing[10,11]. In general, biological ion channels ultimate selectivity is considered stemming from the synergy effect of their sub-nanoscale pore size, specific binding sites, and appropriate charge density[12–14]. These physicochemical features have thus largely inspired the

designing principles for current membranes aiming to achieve similar separation performance. Earlier efforts were first devoted to mimicking the structural characteristics of biological ion channels, constructing uniform sub-nanochannels with graphene oxide (GO)[15,16], metal-organic frameworks (MOFs)[6,17] and perforated polyethylene terephthalate (PET)[18,19]. Due to the insufficient selectivity by size exclusion alone, later studies grafted simple groups (e.g., $–SO_3^-$) into the channel and distinguished cations, particularly those same-valent and similar-sized, via discrepant chemical affinity[5,17,20,21]. Meanwhile, the charge density of the channels has also been tailored to modulate ion selectivity[22–24]. While these designs captured one of biological ion channel feature, to replicate all the three in high precision remain

[1]State Key Laboratory of Pollution Control and Resource Reuse, School of the Environment, Nanjing University, 210023 Nanjing, China. [2]Research Center for Environmental Nanotechnology (ReCENT), Nanjing University, 210023 Nanjing, China. [3]Department of Chemical and Biological Engineering, Monash University, Clayton, VIC 3800, Australia. [4]UQ Dow Centre for Sustainable Engineering Innovation, School of Chemical Engineering, The University of Queensland, St Lucia, QLD 4072, Australia. ✉e-mail: wmzhang@nju.edu.cn; bcpan@nju.edu.cn; xiwang.zhang@uq.edu.au

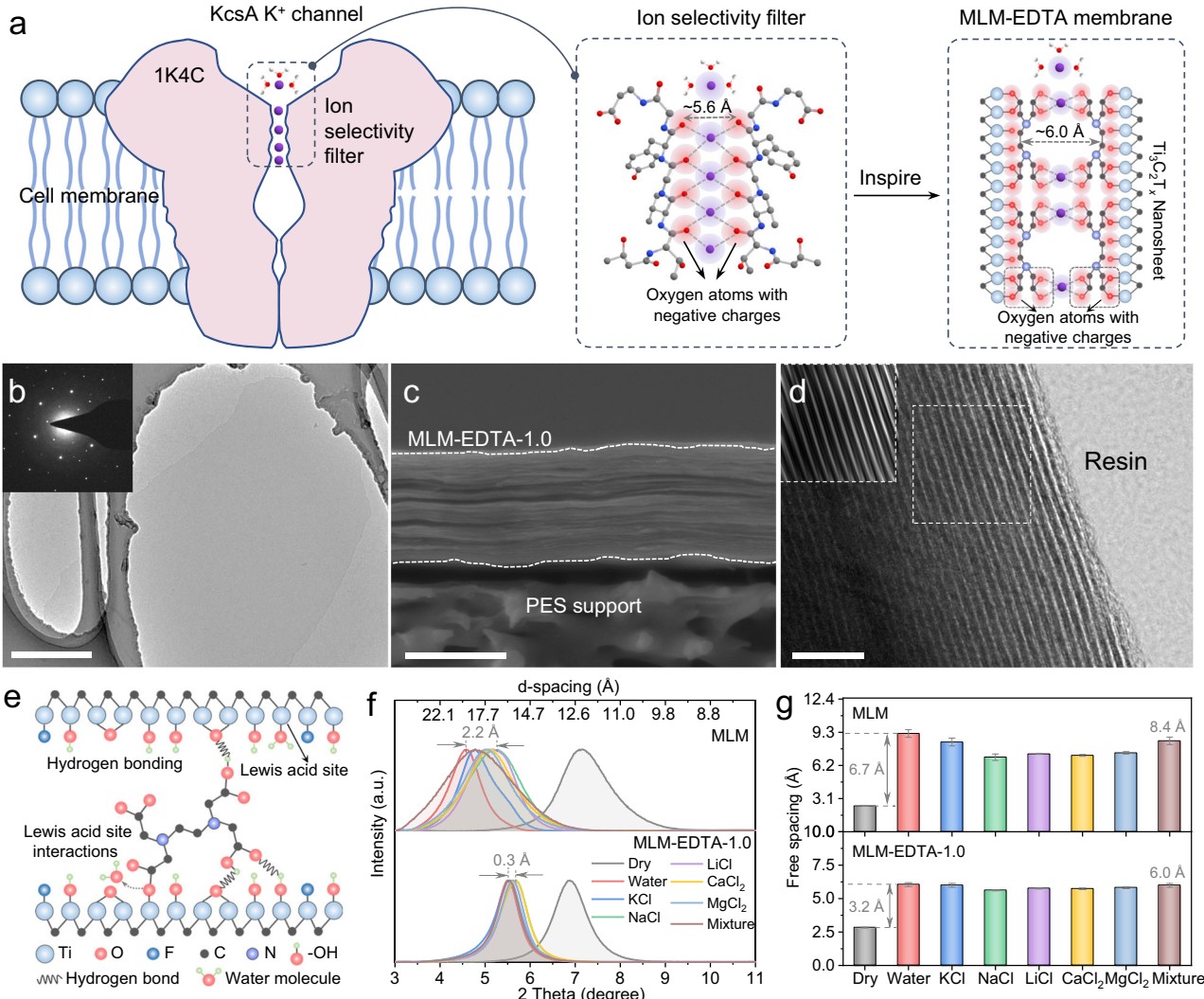

**Fig. 1 | Preparation and characterization of the MLM-EDTA. a** The ion selectivity filter in KcsA K+ channel is a 5.6 Å channel composed of eight negatively charged oxygen atoms (PDB code 1K4C), which inspired us to design an MLM-EDTA with numerous negatively charged oxygen atoms and 6.0 Å two-dimensional channel. **b** TEM image and selected area electron diffraction (SAED) pattern of the Ti₃C₂Tₓ nanosheets. Scale bar, 1 μm. **c** SEM image of the cross-section for MLM-EDTA-1.0. Scale bar, 1 μm. **d** TEM image of the cross-section of the MLM-EDTA-1.0 membrane, insert: the computerized TEM image obtained from the inverse Fast Fourier-Transform (IFFT) and FFT patterns after removing background noise. Scale bar, 10 nm. **e** Schematic diagram of the cross-linking mechanism between Ti₃C₂Tₓ nanosheets and EDTA molecule. **f** Comparison of anti-swelling properties between MLM and MLM-EDTA-1.0 membranes obtained from XRD patterns. The XRD patterns of the wet samples were measured immediately after being immersed in different solutions (DI water, 0.2 M KCl, NaCl, LiCl, CaCl₂, MgCl₂ single salt solutions and mixed salts solution) for 3 h. **g** The free-spacings (see Supplementary Fig. 12 for details) of MLM and MLM-EDTA-1.0 in the dry state and various solutions obtained from the XRD patterns in (**f**). Error bars represent the standard deviation of three measurements of a sample.

grandly challenging. Moreover, the transport and separation mechanisms of ions within sub-nanochannels modulated by local charge density remains elusive and an effective strategy to fabricate high-efficiency mono-/divalent ions selective membranes is yet to be developed. Inspired by KcsA K+ channel with size of ~5.6 Å and homogenous distribution of carbonyl oxygens, herein, MXene nanosheets and EDTA molecules are used as building blocks to construct artificial ion channel membranes with the right channel size, similar binding sites, and tunable charge density (Fig. 1a).

Ti₃C₂Tₓ as a new 2D materials from MXene family is a promising building block for biomimetic channels. With its Ti-C-Ti multi-sublayer structure, Ti₃C₂Tₓ possesses intrinsic rigidity and inter-sheet attraction to allow them assemble into stacked sub-1-nm channels with superior uniformity[25–30]. Besides, the surface of Ti₃C₂Tₓ is terminated by either numerous −O, −OH and −F[31,32] or Lewis acid Ti sites[33,34], which enables the bridging of neighboring nanosheets via covalent and non-covalent bonding to obtain channel size and structural stability comparable to

biochannels[35–37]. MXene membranes have shown excellent thermal stability, mechanical flexibility, antibacterial properties, and the ability for scalable manufacturing, indicating their potential for applications in water treatment[25,27,38,39]. EDTA, on the other side, has two C−N bonds and four carboxyl groups (Fig. 1a), similar to the binding sites (eight partially negatively charged oxygen atoms from main-chain carbonyl or sidechain hydroxyl) in KcsA K+ channel[13]. In addition to interacting with Ti₃C₂Tₓ via its carboxyl groups to control channel size, EDTA molecules have exceptional ability to complex with divalent cations, so able to recognize mono-/divalent cations[40,41]. The biomimetic features of both host (Ti₃C₂Tₓ) and guest (EDTA) materials, and their good compatibility make them a potential combination to produce high-performance artificial ion channel membranes.

A facile method is employed in this study to construct MXene laminar membranes functionalized with EDTA (MLM-EDTA), which have well-aligned, compact and water-stable 2D sub-nanochannels. MLM-EDTA membranes show excellent cation sieving performance, in

particular, a K$^+$/Mg$^{2+}$ selectivity of 121.2 in mixed salt solution. The advanced characterization and theoretical simulation reveal that the cation recognition effect of EDTA and size exclusion play critical roles, both of which lead to a high energy barrier for divalent cations. On this basis, we further prove the positive impact of local charge density around EDTA on K$^+$/Mg$^{2+}$ selectivity by strengthening its ion-recognizing effect via complexion. This study provides new insights to the design and development of artificial ion channel membranes, which are highly demanded in various applications.

## Results and discussion

### Preparation and characterization of the EDTA-modified membranes

Ti$_3$C$_2$T$_x$ nanosheets were prepared by etching away Al layer from parent Ti$_3$AlC$_2$ phase with a minimally intensive layer delamination (MILD, Supplementary Fig. 1) method[32]. As-synthesized nanosheets possessed an average flake size of 1.0 μm and thickness of around 1.5 nm (Supplementary Fig. 2a), close to the theoretical monolayered Ti$_3$C$_2$T$_x$ thickness of 1.0 nm[29,42]. When examined under transmission electron microscopy (TEM), these nanosheets presented flat and ultrathin features, and exhibited single crystal diffraction pattern under selected area electron diffraction (SAED) mode, further confirming their single-layer nature (Fig. 1b and Supplementary Fig. 2b–d). X-ray photoelectron spectroscopy (XPS, Supplementary Fig. 3) indicated two major post-synthesis changes on material chemical composition, including the complete removal of Al, as proved by the disappeared Al 2$p$ Spectra (Supplementary Fig. 3a, b), and the introduction of surface –OH, –O groups and Lewis acid Ti sites, as shown in Ti 2$p$ and O 1$s$ spectra (Supplementary Fig. 3c, d). In the following step of nanosheet modification by EDTA, these active surface sites allow versatile EDTA-Ti$_3$C$_2$T$_x$ connections via both covalent (with Lewis acid Ti[26,40]) and non-covalent bonding (with hydrogen bond)[36] after simply mixing. This led to the uniform distribution of EDTA molecules on Ti$_3$C$_2$T$_x$ nanosheets, as shown by high-angle annular dark-field (HAADF) images and corresponding element mappings (Supplementary Fig. 4). Simultaneously, increasing negative zeta potential (Supplementary Fig. 5) and emerging N 1$s$ spectra in XPS (Supplementary Fig. 6) also imply the successful EDTA grafting onto the nanosheets.

The pre-modified Ti$_3$C$_2$T$_x$ with varying EDTA loading were then assembled into laminar membranes via vacuum-assisted filtration and named by MLM-EDTA-X (X ranging from 0 to 1.5 mg mL$^{-1}$). All membranes illustrated highly aligned cross-section under scanning electron microscope (SEM, Fig. 1c and Supplementary Fig. 7) and TEM (Fig. 1d), and homogeneous EDTA distribution under SEM elemental mapping (Supplementary Fig. 8). Although the increasing EDTA loading posed minor impacts on membrane surface morphology, it clearly contracted the laminar structure and rendered the membrane more compact along the membrane thickness direction, thus eliminating the macroscopic voids intrinsic to pristine membranes[37]. This could be attributed to EDTA-enabled Ti$_3$C$_2$T$_x$ nanosheet crosslinking via Ti–COO$^-$ covalent bonding and –COOH–OH and –COOH–O hydrogen bonding, as revealed by XPS (Supplementary Fig. 9a, b) and Fourier-transform infrared spectroscopy (FT-IR, Supplementary Fig. 9c), respectively. Correspondingly, MLM-EDTA membranes displayed much smaller and more stable channel spacing in various aqueous environments (Fig. 1f). It is noteworthy that the full width at the half maximum (FWHM) of the XRD peaks for membranes in saline solution decreased dramatically after modified with EDTA, indicating the Ti$_3$C$_2$T$_x$ nanosheets become more compact and aligned due to the interactions between EDTA and nanosheets[43,44]. While the spacing of MLM increased from 2.5 Å to up to 9.2 Å when they were transferred from dry to aquatic condition, that of MLM-EDTA-1.0 only increased from 2.8 to 6.0 Å (Fig. 1g). The significant improvement in the anti-swelling property of MLM-EDTA in saline solution is attributed to the extensive covalent and hydrogen bonding interactions between EDTA

molecules and Ti$_3$C$_2$T$_x$ nanosheets, which bridge neighboring nanosheets together. Such "glue" effect of EDTA not only addressed the undesired swelling phenomenon universally existing in most 2D channels, but also ensured a fixed channel size comparable to that of KcsA K$^+$ channel (5.6 Å), which facilitates the design of bioinspired ion channels. Moreover, we found the d-spacing of MLM and MLM-EDTA membranes obtained from XRD patterns (Supplementary Fig. 10) agreeing well with cross-sectional TEM image results (Supplementary Fig. 11). This suggested the high accuracy of our channel spacing measurement through XRD spectra.

### Ion separation performance of MLM-EDTA membranes

The ion separating capability of MLM-EDTA membranes compared to pristine MLM membranes were evaluated by measuring the permeation rates of mixed alkali and alkaline ions (K$^+$, Na$^+$, Li$^+$, Ca$^{2+}$, Mg$^{2+}$) using a homemade U-shaped device (Fig. 2a). MLM membrane demonstrated a monovalent/divalent (K$^+$/Mg$^{2+}$) selectivity of around only 10 and hardly any selectivity between monovalent cations (Fig. 2b). This limited selectivity could be attributed to the relatively large channel size (8.4 Å, Fig. 1g and Supplementary Fig. 13) that barely distinguished between ions via size exclusion (8.24 Å for Ca$^{2+}$, 8.56 Å for Mg$^{2+}$, Supplementary Table 1). On the contrary, MLM-EDTA membranes slightly affected K$^+$ while pronouncedly impeded the transport of the rest ions, thus enhancing K$^+$/M$^{n+}$ selectivity by a great extent. In particular, MLM-EDTA-1.5 achieved much improved K$^+$/Li$^+$ and K$^+$/Mg$^{2+}$ selectivity of 5.4 and 121.2, respectively, compared to 1.4 and 10.2 for MLM (Fig. 2c and Supplementary Fig. 14a). Considering the reduced channel size (below 6.0 Å Supplementary Fig. 13), size induced ion partial dehydration should play a significant role in the enhancement. To adapt into the nanochannels, these hydrated ions would strip off a fraction of their hydration shell, the required energy of which is in positive correlation of their respective full hydration energy[15,22,45–47]. Since such cation hydration energy follows K$^+$ (−295 kJ mol$^{-1}$) < Na$^+$ (−365 kJ mol$^{-1}$) < Li$^+$ (−475 kJ mol$^{-1}$) ≪ Ca$^{2+}$ (−1505 kJ mol$^{-1}$) < Mg$^{2+}$ (−1830 kJ mol$^{-1}$) (Supplementary Table 1), the transport energy barrier confronted by the different ions at MLM-EDTA channel entry should remain the similar order, thus justifying our observed ion permeation rate of K$^+$> Na$^+$> Li$^+$ ≫ Ca$^{2+}$> Mg$^{2+}$. Moreover, the high separation performance was well maintained over long-term and cycling tests in water, indicating the anti-swelling advantage of MLM-EDTA and its real-world application potentials (Supplementary Figs. 15 and 16).

It is interesting to note that increasing EDTA loading into the membrane from 0.25 to 1.0 mg mL$^{-1}$, though no more narrowing the channels, further improved the K$^+$/Mg$^{2+}$ selectivity (Supplementary Figs. 13, 14b). This implies another role of EDTA molecules in mediating cation separation in addition to channel size regulation. To this end, we recalled the different connecting patterns of EDTA with cations. With carboxyl groups of EDTA molecules, it can dissociate to impart the channel negative charges to attract alkali cations via ionic bond (Fig. 2d). However, the deprotonated carboxyl (–COO$^-$), together with two electronegative N atoms, can serve as ligands to coordinate with alkaline earth and transitional metal ions, forming much more chemically stable complex (Fig. 2e). To prove the ionic recognizing role of EDTA in membranes, we then implemented density functional theory (DFT) calculations to study EDTA-M$^{n+}$ combinations (Supplementary Fig. 17). As expected, DFT revealed its much higher binding energy with Mg$^{2+}$ (−3.50 eV) and Ca$^{2+}$ (−2.42 eV) than with monovalent Li$^+$ (−1.49 eV), Na$^+$ (−1.20 eV), and K$^+$ (−0.98 eV). This led to stronger Mg$^{2+}$ affinity with anchored EDTA and thereby greater transporting energy barrier through the channels (Supplementary Fig. 18)[5,17,20]. Meanwhile, the differential affinity of EDTA molecules towards monovalent cations endows MLM-EDTA with superior capability for separating monovalent ions, achieving a K$^+$/Li$^+$ selectivity up to 5.4 (Supplementary Fig. 14a). This suggests that rational design of affinity groups inside sub-nanochannels may lead to highly selective separation of

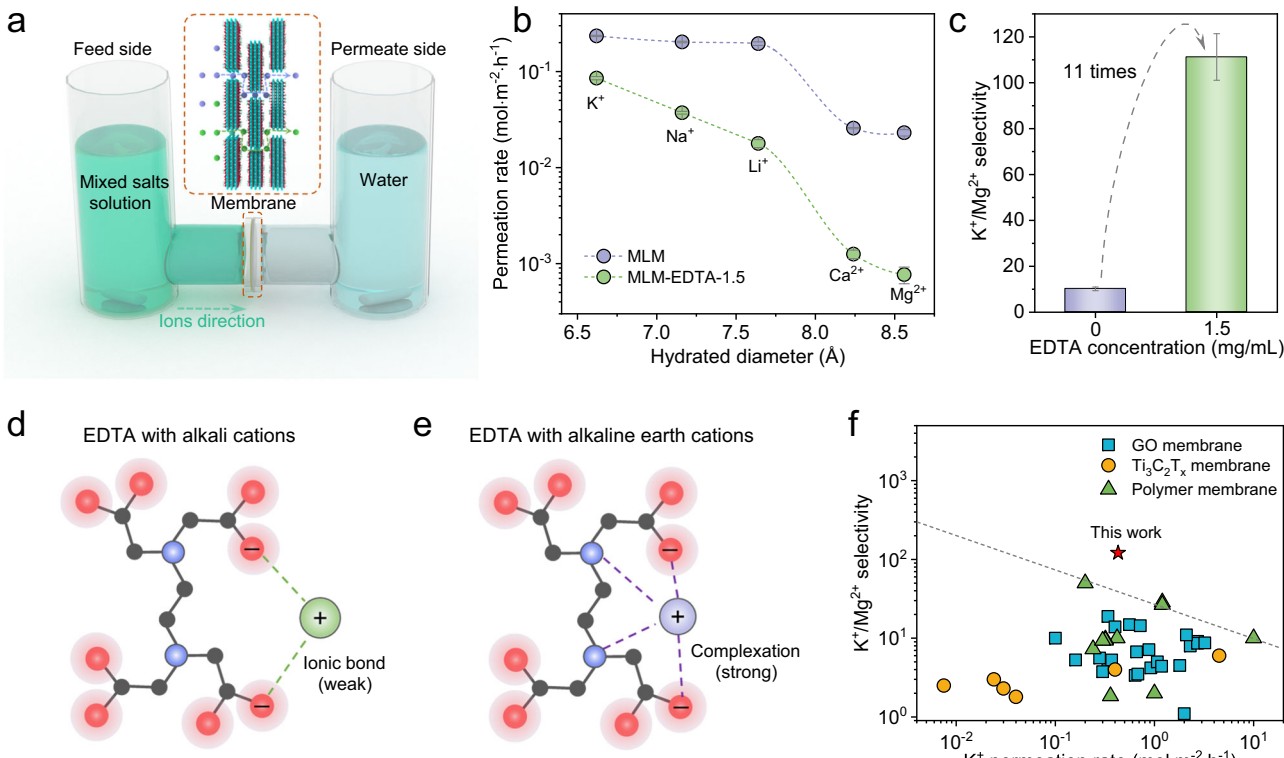

**Fig. 2 | Ion sieving performance of MLM-EDTA. a** Schematic of the homemade U-shaped device used for ion permeation measurements. An encapsulated membrane (see Supplementary Fig. 39 for details) was fixed between feed and permeate chambers, magnetic rotors on both sides to prevent the effects of concentration polarization on the membrane surface. **b** Ions permeation rates through MLM and MLM-EDTA-1.5. **c** EDTA dependent K$^+$/Mg$^{2+}$ selectivity of MLM-EDTA membrane. **d** Schematic diagram of the interaction between EDTA molecules and alkali cations. **e** Schematic diagram of the interaction between EDTA molecules and alkaline earth cations. **f** Performance comparison of K$^+$/Mg$^{2+}$ selectivity vs K$^+$ permeation rates for various membranes under similar testing conditions (see Supplementary Table 2 for details). Error bars represent the standard deviation of three measurements of a sample.

monovalent ions. Notably, MLM-EDTA achieving excellent K$^+$/Mg$^{2+}$ selectivity of 121.2 using mixed salt solution as the feed, which outperforms reported previously membranes under similar measuring conditions and transcends the current upper limit (Fig. 2f and Supplementary Table 2).

**Charge density-dependent ion separation**

To verify the ion-recognizing role of EDTA-decorated sites in MXene 2D channels, we managed to alter their local charge density via adjusting the operating pH. With gradual pH increase, the local charge density (calculated with surface zeta potential, Eq. (2)) in all MLM-EDTA membranes showed a rise without affecting channel size (Supplementary Fig. 19 and Fig. 20, Eq. (2)). Particularly for MLM-EDTA-1.5, the transition of pH from 2.5 to 8.0 caused charge density to change from −7.0 to −22.0 mC m$^{-2}$, a more remarkable increase than that for pristine MLM membranes (Fig. 3a). This implied that the enriched charge density throughout MLM-EDTA not only stemmed from the deprotonation degree of Ti$_3$C$_2$T$_x$-affiliated hydroxyl (Supplementary Fig. 21), but also that of the carboxyl from the anchored EDTA (Fig. 3b). Correspondingly, such charge density increases localized around EDTA sites, which greatly boosts K$^+$/Mg$^{2+}$ selectivity for MLM-EDTA-1.5 from 53.8 to 112.5 while the selectivity of pristine membranes only shows a minor improvement from 6.0 to 9.5 over the tested pH range (Fig. 3c). It is worth noting that the charge density and the K/M$^{n+}$ (M$^{n+}$ are Na$^+$, Li$^+$, Ca$^{2+}$, and Mg$^{2+}$, respectively) selectivity were linearly correlated (Fig. 3d and Supplementary Fig. 22). A more detailed analysis on ions transporting resistance (reciprocal of ion permeation rates, $1/P_i$) revealed two critical trends explaining the exceptional selectivity (Fig. 3e and Supplementary Figs. 24 and 25). First, compared to pristine MLM membranes, MLM-EDTA membranes posed much higher Mg$^{2+}$ transporting

resistance, while K$^+$ encountered substantially smaller impediment in both membranes. Second, this Mg$^{2+}$-targeted resistance linearly soared with denser channel charge while K$^+$ still remained largely unaffected.

These trends well reflected and confirmed the ability of EDTA sites in the membrane to distinguish cations of varied valence based on connection modes. In acidic pH range, the overall low charge density allowed limited electrostatic attraction of K$^+$ to the channels, but does not prevent Mg$^{2+}$ from partly coordinating with EDTA via its electronegative N atoms. The inherent binding strength discrepancy between these covalent bonds and non-covalent interactions thereby sets a decent benchmark for K$^+$/Mg$^{2+}$ selectivity. When operating environment becomes more alkaline, higher charge density due to functionality deprotonation intensifies the K$^+$-channel attraction by a small margin. At the same time, the deprotonation turns −COOH into −O=C−O$^-$, the negative charge of which can hybrid with C=O π electron to form delocalized π cloud. Such electron-donating ligands, up to 4 in an EDTA molecule, will further engage in the complexion with Mg$^{2+}$ to ultimately form octahedral chelates that are substantially stronger than K$^+$-channel attraction. This analysis is readily supported by the experimental comparison of cation transmembrane energy barrier when membrane charge density elevates from −7.0 to −22.0 mC m$^{-2}$ (Fig. 3f and Supplementary Fig. 27). While the barrier for both ions enlarges within this range, their barrier difference at −22.0 mC m$^{-2}$ reaches 5.78 Kcal mol$^{-1}$, almost doubled from 3.09 Kcal mol$^{-1}$ at −7.0 mC m$^{-2}$. Such difference is also corroborated by DFT calculations that quantify the interaction of ions with EDTA molecules of varying deprotonation degree, denoted by the number of negatively charged carboxylates (Fig. 3g and Supplementary Fig. 28). It is shown that the successive dissociation of protons from carboxyl kept widening the ion-EDTA adsorption energy gap, showing far firmer

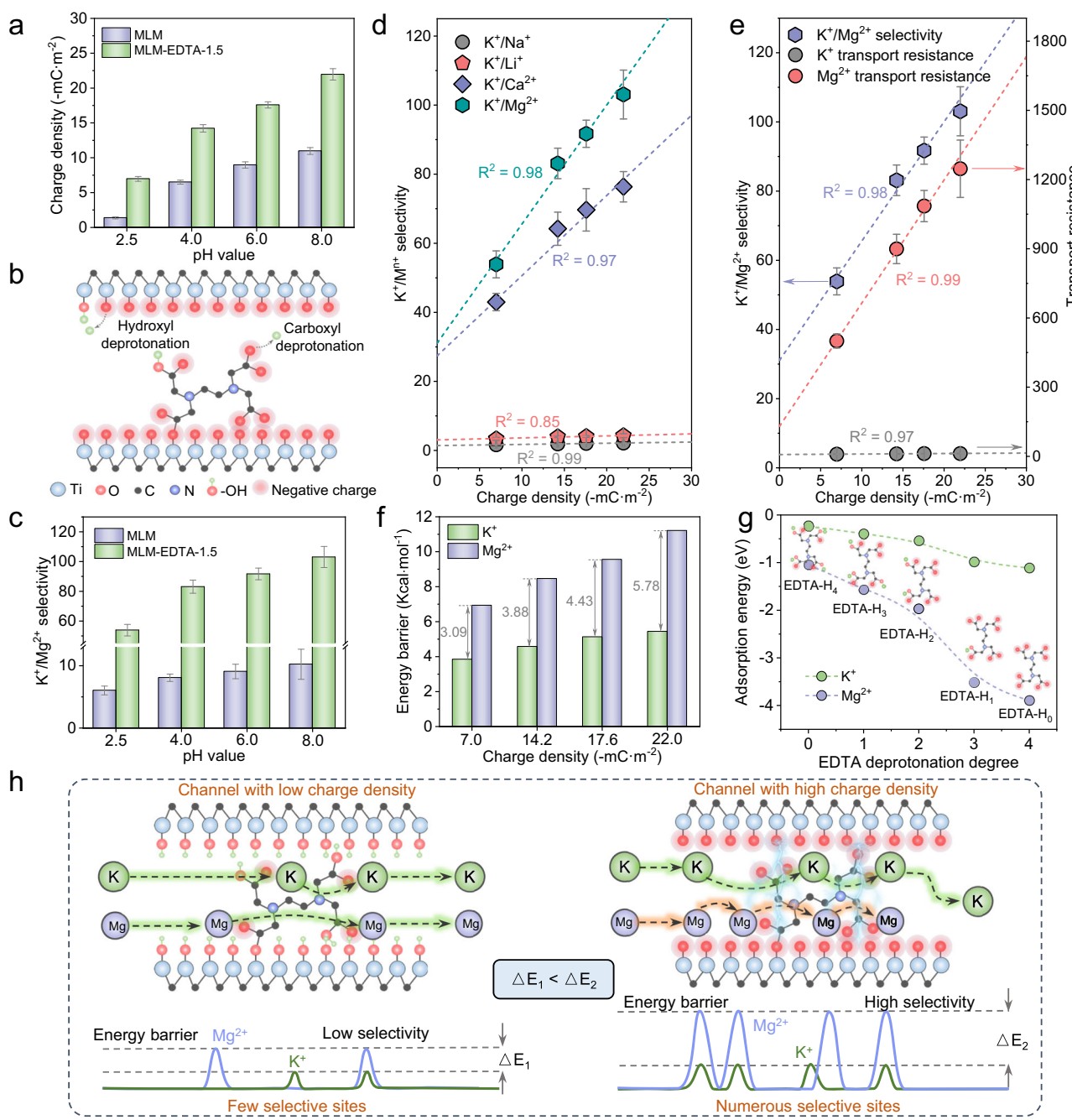

**Fig. 3 | Charge density-dependent ion separation. a** Charge densities of MLM and MLM-EDTA-1.5 membranes at different pH, calculated by the membranes surface zeta potential (Eq. (2)). **b** Schematic diagram of the main sources of negative charges in MLM-EDTA. **c** pH-dependent K⁺/Mg²⁺ selectivity of MLM and MLM-EDTA-1.5. **d** Relationship between charge density and K⁺/Mⁿ⁺ selectivity for MLM-EDTA-1.5 membrane. **e** Relationship between K⁺, Mg²⁺ transport resistance (reciprocal of ion permeation rates, $1/P_i$) and K⁺/Mg²⁺ selectivity with charge density as independent variable for MLM-EDTA-1.5 membrane. **f** The transport energy barrier (obtained from Arrhenius plots, Supplementary Fig. 27) for K⁺ and Mg²⁺ diffusion through MLM-EDTA-1.5 membrane in a 0.2 M mixed salt solution at different charge densities. **g** The DFT calculated adsorption energy for K⁺ and Mg²⁺ on EDTA molecules with different negatively charged oxygen atoms. **h**, Schematic illustration of the possible mechanism for charge density enhancing the K⁺/Mg²⁺ selectivity in intra-membrane diffusion process. Error bars represent the standard deviation of three measurements of a sample.

affinity to Mg²⁺ to realize its recognition from the surrounding K⁺. These evidences consolidate the role of the anchored EDTA molecules as mono-/divalent cations selective sites, especially in higher charged density (Fig. 3h). We also notice that, for the Mg²⁺ transport resistance data, the intersection of linear fits with y-axis is not zero, indicating other causes of transport resistance including size effect[22,46,47] and hydrogen bonding effect[48]. To interpret the contribution of other effects (except charge effect) to the resistance for Mg²⁺ transport, we converted the intersection of linear fits with y-axis into the

contribution of other effects, and found the highest contribution of other effects by only 8.4% for MLM-EDTA-1.5 (Supplementary Fig. 26). This indicates that the local charge density in MLM-EDTA sub-nano-channel plays significant roles in cation separation.

## Ion transport behaviors in 2D channels of MLM-EDTA membranes

Considering the above-mentioned multirole of EDTA in 2D MXene channels, we then studied how the interplay of its various effects

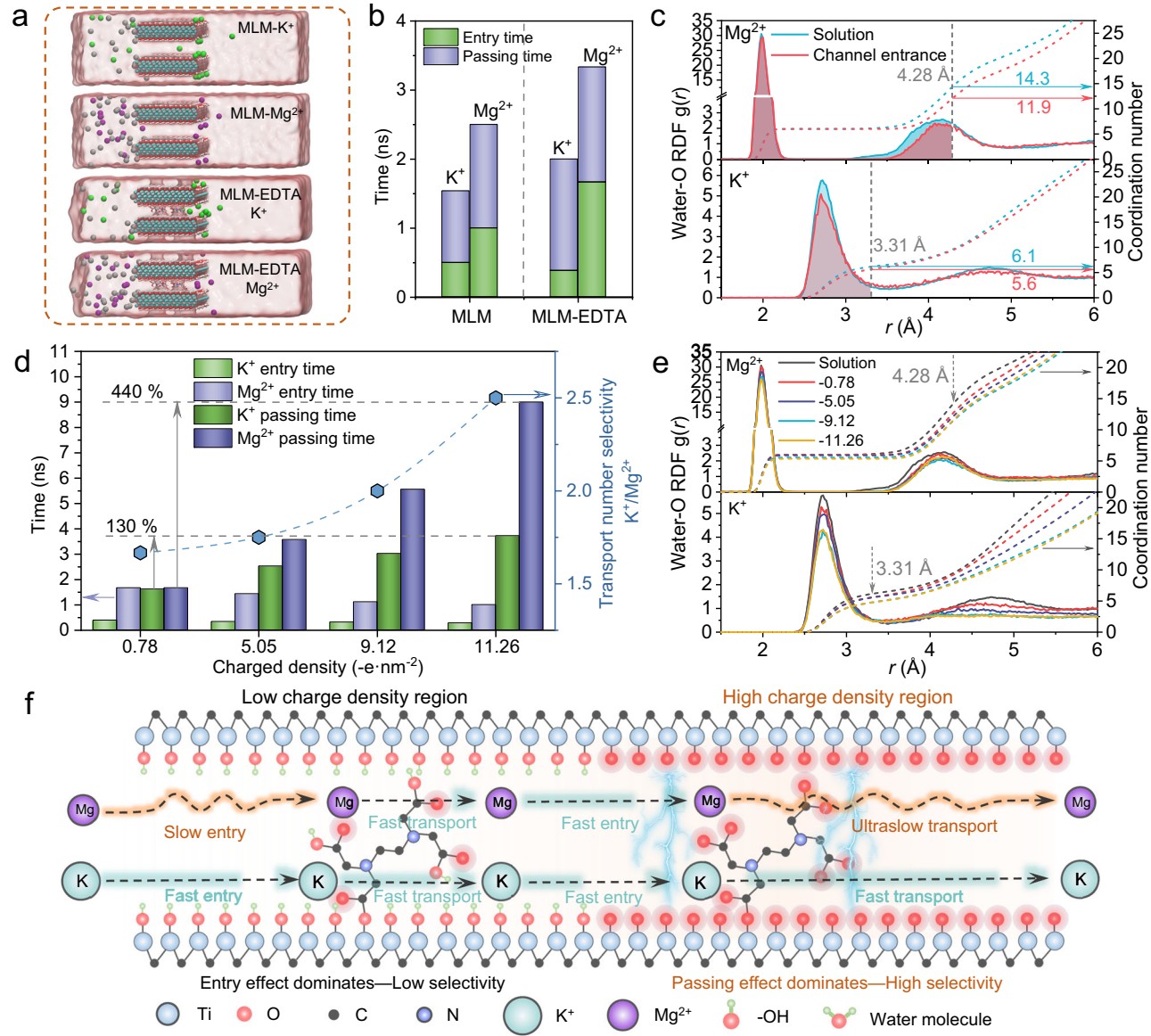

**Fig. 4 | MD simulation of ion transport across MLM-EDTA. a** Snapshots of MD simulation for K⁺ and Mg²⁺ passing the MLM and MLM-EDTA systems at 20 ns. K⁺, Mg²⁺, and Cl⁻ ions are in green, purple and silver, respectively. **b** Average entry time and passing time of K⁺ and Mg²⁺ passing the MLM and MLM-EDTA systems. **c** Radial distribution function (RDF) of oxygen in water molecules around K⁺ and Mg²⁺ locating at the solution and MLM-EDTA channel entrance. The valid range of the data was taken from 0 to the hydration radius of the ions, K⁺ and Mg²⁺ are 3.31 Å and 4.28 Å (Supplementary Table 1), respectively. **d** Average entry time and passing time of K⁺ and Mg²⁺ passing MLM-EDTA systems with different number of negatively charged oxygen atoms. **e** RDF of oxygen in water molecules around K⁺ and Mg²⁺ locating at the solution and inside of MLM-EDTA channel, the channel with different charge densities. **f** Schematic illustration of the possible mechanism for the role of charge density on cations transport and separation in sub-nanochannels.

regulates overall transmembrane ion transport under different conditions with molecular dynamics (MD) simulations (Fig. 4a and Supplementary Fig. 30)[18,22,45,46]. Figure 4b shows that K⁺ passes the pristine channel (*d*-spacing 8.4 Å) in slightly less time than Mg²⁺, and this time gap is enlarged by 20% in MLM-EDTA channels (*d*-spacing 6.0 Å), in accordance with our experimental results. To quantify such differences in details, we broke down the transport process into two critical steps, including ion entry and ion passing. Compared to that in pristine channel, the entry time (time interval between two adjacent ions to enter the channel) in the EDTA-decorated MXene channel prolonged for Mg²⁺ due to reduced channel size, but unexpectedly shortened for K⁺. The counter-intuitive phenomenon is largely the competing result of size repulsion and electrostatic attraction caused by EDTA at the same time. Radial distribution function (RDF, Fig. 4c) implies the drastically discrepant dehydration degree of Mg²⁺ and K⁺ at MLM-EDTA

channel entry, losing 2.4 (14.3–11.9) and 0.5 (6.1–5.6) water molecules from their respective hydration shells, whereas no dehydration at MLM channel entry (Supplementary Fig. 31). While the strengthened electrostatic attraction in EDTA-decorated channels is still too weak to offset the huge dehydration hindrance of Mg²⁺, it can overcome that of K⁺ to slightly reduce its entry time. Meanwhile, the passing time (time required for an ion to travel through the channel) for both ions is similar in EDTA-decorated channels, identifying size-related ion entry as the transport rate-limiting step and the origin of K⁺/Mg²⁺ selectivity at relatively low charge density of −0.78 e nm⁻² (Supplementary Fig. 32). However, when the charge density is gradually tuned up from −0.78 e nm⁻² to −11.26 e nm⁻², the Mg²⁺-channel electrostatic attraction becomes sufficient to surpass dehydration energy barrier and, like in the case of K⁺, leads to deceased Mg²⁺ entry time (Fig. 4d and Supplementary Fig. 33). By contrast, the passing time of Mg²⁺ experiences a

massive increase by 440% whereas that of $K^+$ by only 130%, resulting in greatly enhanced $K^+/Mg^{2+}$ separation. Further considering the continuously higher contribution of passing time to total transport time (Supplementary Fig. 34), we attributed the enhancement to the factors that impede $Mg^{2+}$ permeation in high charge density, such as the strong $Mg^{2+}$-EDTA complexation discussed in the above DFT.

Noteworthily, we discovered that for partially dehydrated cations already in the channel, their transport was subject to additional regulation in a more charged environment by undergoing a second-time ion dehydration[47,49]. With the charge density raising from −0.78 to −11.26 e $nm^{-2}$, the hydration number of $K^+$ and $Mg^{2+}$ both decreased (Fig. 4e), RDF of hydrogen atoms revealed the establishment of more hydrogen bond between the hydration shells of cations, particularly $Mg^{2+}$, and those deprotonated hydroxyl groups $-O^-$ on channel walls (Supplementary Fig. 35). Not only will the extra hydrogen bond cause more $Mg^{2+}$-channel friction, but also the re-dehydrated $Mg^{2+}$ can facilitate its bonding with EDTA, both of which would further augment its transporting resistance and consequently maximize $K^+/Mg^{2+}$ selectivity (Supplementary Fig. 36). Meanwhile, an entry-passing dominant transport mechanism in 2D sub-nanochannels is proposed, the cations selectivity is dominated by the entry effects in a low charge density channel, whereas in the high charge density channel, the cation selectivity is dominated by the passing effects (Fig. 4f).

In summary, inspired by KcsA $K^+$ channel, MLM-EDTA membranes with biomimetic channel size, binding sites and tunable charge density have been successfully prepared. MLM-EDTA membranes have highly alignment, compactness and water-stable 2D sub-nanochannels and achieved ultrahigh, pH tunable mono-/bivalent cations selectivity and permeability in mixed salt solutions (KCl, NaCl, LiCl, $CaCl_2$, $MgCl_2$) permeation conditions. Our experiments and simulations indicate that the cation recognition effect of EDTA and partial dehydration effects plays critical roles for cations selective sieving and increasing the local charge density within the sub-nanochannel significantly improves cation selectivity. The cation selectivity improved by charge density is mainly attributed to the remarkable enhancement of interatomic affinity and hydrogen bonding between hydrated bivalent cations and channel. The cations selectivity might be dominated by the entry effects in a low charge density channel, whereas in a high charge density channel, the cation selectivity is dominated by the passing effects. The biomimetic MLM-EDTA demonstrated here provide an attractive strategy to fabricate high-performance ion separation membranes learning from biological ion channels to rationally design the affinity groups and local charge density for applications such as resource recovery, clean water production and power generation.

## Methods

### Chemicals and reagents
The bulk material MAX ($Ti_3AlC_2$, 400 mesh) was purchased from Kaixi Tech, China. LiF (≥99%), EDTA-2Na (≥98%), HCl (12 M), KCl, NaCl, LiCl, $CaCl_2$, and $MgCl_2$ were purchased from Sinopharm Chemical Reagent Co., Ltd, China. Polyethersulfone (PES) substrate (0.22 μm) was purchased from CNW ANPEL Laboratory Technologies Inc., Shanghai, China. AAO disc filters (100-nm pore, 25-mm diameter) were purchased from GE Healthcare Whatman. All the chemicals were used as received without further purification. Deionized water was produced by an ULUPURE ultrapure water system (ULUPURE, China).

### Preparation of the MXene ($Ti_3C_2T_x$) nanosheets
$Ti_3C_2T_x$ nanosheets were prepared by improved the minimally intensive layer delamination (MILD) method by selective etching Al atom layer from $Ti_3AlC_2$ using HCl and LiF mixed aqueous. Typically, LiF (1.0 g) was mixed with HCl (20 mL, 9 M) in a Teflon vessel by stirring for 3 min. Subsequently, $Ti_3AlC_2$ (1.0 g) powder was gradually added into the etchant solution, followed by continual stirring (300 rpm) for 24 h at 37 °C. After that, the obtained acidic product was transferred into a

50 mL centrifuge tube, followed by centrifugation (5 min, RCF of 1238×$g$). Poured 40 mL deionized (DI) water into the obtained product, and divided it into two parts evenly after mixing with a vortex mixer (IKA S25, Germany). Then, the resulting dispersion was washed with DI water by repeated centrifugation (CT14RD, Techcomp Co., Ltd, China) at RCF of 1238 to 4951×$g$ for 5 min per cycle until self-delamination occurred at a supernatant pH of ~6. The resulting sediment was then diluted with DI water and ultrasonicated (110 W) for 30 min in an ice water bath under Ar flow. Finally, the monolayer $Ti_3C_2T_x$ nanosheets dispersion was obtained by centrifugation at RCF of 2200×$g$ for 60 min. The as-obtained $Ti_3C_2T_x$ dispersion usually has a concentration around 2 mg $mL^{-1}$. Pump Ar gas (more than 15 min) into the dispersion before storing it in low temperature (4 °C).

The concentration of $Ti_3C_2T_x$ nanosheets was measured by membrane mass difference subtraction. Briefly, deposited $Ti_3C_2T_x$ nanosheets on PES substrate (0.22 μm), record the mass of the PES substrate before $Ti_3C_2T_x$ deposition and after deposition. The $Ti_3C_2T_x$-PES membrane simples were vacuum drying at 60 °C for 6 h. $Ti_3C_2T_x$ nanosheets concentration, $C_T$ (mg $mL^{-1}$), is calculated as follow:

$$C_T = \frac{M_1 - M_0}{V} \tag{1}$$

where $M_1$ (mg) is the PES substrate mass after $Ti_3C_2T_x$ deposition, $M_0$ (mg) is the PES substrate mass before $Ti_3C_2T_x$ deposition, $V$ (mL) is the volume of $Ti_3C_2T_x$ nanosheets dispersion. The concentration measurement was repeated three times. Diluted the obtained dispersion concentration to 1.0 mg $mL^{-1}$ for use.

### Fabrication of the MXene and MXene-EDTA membrane
Prepared EDTA-2Na solutions with concentrations of 0, 0.25, 0.5, 1.0, and 1.5 mg $mL^{-1}$, respectively. Added the $Ti_3C_2T_x$ dispersion (1.0 mg $mL^{-1}$) to 50 mL EDTA-2Na solution, followed by continuous stirring at room temperature for 6 h. Then poured the $Ti_3C_2T_x$-EDTA mixture solution into a suction filtration device sandwiched with a PES membrane (0.22 μm, 47 mm). After standing for 10 min, the mixture solution was filtered with a pressure of 0.3 bar for 10 min, then the pressure was increased to 1 bar with a pressure regulating valve and keep filtration. After there was no solution on the membrane surface, filtration was continued for 10 min to remove the residual solution in the interlayer spacing. The obtained membrane was vacuum-dried at 80 °C for 6 h. After naturally cooling down to the room temperature, the membrane was washed with DI water for 3 times and then immersed in DI water for 3 h to remove uncrosslinked EDTA molecules. By the difference in EDTA-2Na concentration, the synthesized membranes were named as MLM, MLM-EDTA-0.25, MLM-EDTA-0.5, MLM-EDTA-1.0, MLM-EDTA-1.5, respectively.

### Characterizations
XRD patterns were carried out with an X-ray diffractometer (XRD, Bruker D8 advance, Germany) equipped with a Cu sealed tube ($\lambda$ = 1.54178 Å) at 40 kV and 40 mA at room temperature, the samples were scanned at 0.01 degree per step. The morphologies and cross-sections of the membranes were observed by an environmental scanning electron microscope (ESEM, Quanta 250 FEG, USA) equipped with energy-dispersive X-Ray spectroscopy (EDS) under 5 kV (membrane) or 30 kV ($Ti_3C_2T_x$ nanosheet deposited on AAO substrate) with a spot size of 2.5. All samples were coated with 5 nm gold prior to SEM examination, deduct 5 nm thick gold during EDS analysis. Atomic force microscopy (AFM, Bruker ICON2-SYS, USA) was used to characterize the morphology and thickness of the nanosheets, nanosheet dispersions were spin-coated on mica discs (Diameter of 12 mm, TED PELLA, USA). Fourier-transform infrared spectroscopy was recorded on a Nicolet Instrument (FT-IR, Nicolet iS5, USA) equipment using a KBr

pellet in the range of 400–4000 cm$^{-1}$. X-ray photoelectron spectroscopy (XPS, PHI 5000 VersaProbe-III, Japan) was used to investigated the chemical constitution, and an Al Kα (150 W) X-ray source at a chamber was used to excite photoelectrons. The obtained XPS spectra was processed in XPS peak (Version 4.1) for fitting peaks. Zeta potential of $Ti_3C_2T_x$ nanosheets dispersion measurements were carried out in aqueous solutions using a Zetasizer (Malvern ZS90, UK), and the simple concentration was 100 mg L$^{-1}$, the pH values were adjusted by 0.1 M and 0.05 M HCl and NaOH. Transmission electron microscopy (TEM) and elemental mapping images were obtained using a transmission electron microscope (TEM, Tecnai F20, FEI, USA), operating at voltage of 200 kV. To obtain the membrane cross-sections TEM images, the membrane was first cut into 2 × 8 mm rectangle by blade, then used Epon812 resin for embedding. The embedded membrane samples were processed by an ultrathin microtome (Leica, EM UC 7, Germany, Supplementary Fig. 38), and finally an ultrathin resin sheet exposing the section of the membrane samples was obtained, with a thickness about 50 nm. Membrane surface zeta potential measurements were carried out using a potential analyzer (SurPASS, Anton Paar, Austria). Membranes were cut into 1 × 2 cm squares and taped on the measuring cell using a double-sided tape. Data were collected for two cycles at each measuring point.

## Membrane surface charge density calculation

The membrane charge density (σ, C m$^{-2}$) can be calculated using an Gouy–Chapman equation[48]:

$$\sigma = -\varepsilon\kappa\xi\frac{\sinh\left(\frac{F\xi}{2RT}\right)}{\frac{F\xi}{2RT}} \tag{2}$$

$$\kappa^{-1} = \left(\frac{\varepsilon RT}{2F^2C_0}\right)^{1/2} \tag{3}$$

where $\varepsilon$ ($6.933 \times 10^{-10}$ F m$^{-1}$) is permittivity, $\kappa^{-1}$ (m) is Debye length, $\xi$ (V) is membrane zeta potential, $F$ (96485 C mol$^{-1}$) is Faraday constant, $R$ (8.3145 J mol$^{-1}$ K$^{-1}$) is gas constant, $C_0$ (mol m$^{-3}$) is the concentration of the electrolyte. Here, we assume that channel charge density equals that of the membrane surface.

## Ion permeation measurement

A homemade U-shaped diffusion device with two 30 mL chambers was used for membrane ion permeation measurement (Fig. 2a). The membrane was sealed between two pieces of perforated (diameter of 1.6 cm) aluminum tape (Supplementary Fig. 39), then sandwiching the membrane in the middle of U-shaped diffusion device. The feed side contained five mixed salts (KCl, NaCl, LiCl, CaCl$_2$, MgCl$_2$), each salt concentration is 0.2 M, and the permeate side contained DI water, unless otherwise specified. MXene layer faced the feed side, and the PES substrate faced the permeate side. To avoid concentration polarization effect, two chambers were continuously magnetic stirred during the measure period. During measurements of the reported pH dependences, we adjusted both the feed and permeate solution pH using HCl (0.1 M and 0.05 M), the sequence of pH value was always from high to low. During measurements of the reported concentration dependences, the sequence of used solutions concentration was always from low to high. During measurements of the reported temperature dependences, the measurement was carried out in a constant temperature water bath with magnetic stirring. Once a measurement serie was completed, the device was thoroughly washed with DI water to remove residual ions. This procedure was repeated until the DI water added in each chamber showed the conductance characteristic for pure DI water. After the cleaning, the membrane was stored in pure DI water before used for next measurement. An inductively coupled plasma optical emission spectrometry (ICP-OES, iCAP 7000 series, USA) spectrometer was used for determining of ion concentrations in permeates.

The ion permeation rate $P_i$ (mol m$^{-2}$ h$^{-1}$) calculation is given by:

$$P_i = \frac{(C_1 - C_0) \cdot V}{A \cdot t} \tag{4}$$

where $C_1$ (mol L$^{-1}$) is the permeate side current ion concentration, $C_0$ (mol L$^{-1}$) is the permeate side initial ion concentration (the concentration after 1 h of steady operation), $V$ (L) is the volume of permeate side solution, $A$ is the effective membrane area ($2.01 \times 10^{-4}$ m$^2$), $t$ (h) is the diffusion time.

Membrane ion selectivity, $S$, is calculated as follows:

$$S = \frac{P_{i1}}{C_{i1}} \Big/ \frac{P_{i2}}{C_{i2}} \tag{5}$$

where $P_{i1}$ (mol m$^{-2}$ h$^{-1}$) is the permeation rate of ion 1, and $C_{i1}$ (mol L$^{-1}$) is the concentration of ion 1 in the feed solution (mol L$^{-1}$).

## Ion transport energy barriers measurement

The measurement was carried out in a constant temperature water bath with magnetic stirring, and the temperatures were set to 20, 25, 30, 35, and 40 °C, respectively. The energy barrier ($E_a$) for ion across the membrane can be calculated using an Arrhenius-type equation[6,50,51]:

$$\ln(P_i) = \ln(\alpha) - \left(\frac{E_a}{R} \cdot \frac{1}{T}\right) \tag{6}$$

where α is a pre-exponential factor and $R$ ($1.985 \times 10^{-3}$ kcal mol$^{-1}$ K$^{-1}$) is the gas constant, $T$ (K) is temperature, and $E_a$ (kcal mol$^{-1}$) is the energy barrier. Creation of an Arrhenius plot of the natural log of ion permeation rate ($P_i$) at each reciprocal of the absolute temperature. Then, the determination of the slope of the Arrhenius plot, which is related to the energy barrier divided by the gas constant $R$.

## Density functional theory (DFT) calculation

The calculation of the adsorption energy between the EDTA molecule and the cations were carried out in the Gaussian 09 software package. For the adsorption of EDTA molecules to five cations (K$^+$, Na$^+$, Li$^+$, Ca$^{2+}$, and Mg$^{2+}$), considering that the pH value during the performance measurements were mainly around 8 (unless otherwise specified), the form of EDTA molecule was H$_1$Y$^{-3}$ (see Supplementary Fig. 37 for the EDTA distribution curves at different pH). To simulate the transport and separation behavior of ions under different charge densities in the channel, we carried out the calculation for the adsorption energy of cations and EDTA molecules with different numbers of protons (Supplementary Fig. 28). Structural optimization and frequency analysis commands were executed for each configuration by the density functional theory with the UB3LYP functional in the DEF2SVP unit, and considered the solvation effect of water. The convergence thresholds of maximum force, maximum displacement and energy tolerance were $4.5 \times 10^{-4}$ eV Å$^{-1}$, $1.8 \times 10^{-3}$ Å and $1 \times 10^{-6}$ eV, respectively. The sum of electronic and zero-point energies as the steady-state energy of the system.

The binding energy is calculated as follows:

$$E_{ads} = E_{EDTA-M} - (E_{EDTA} + E_M) \tag{7}$$

where $E_{ads}$ (eV) is adsorption energy, $E_{EDTA-M}$ (eV) is system steady state energy of EDTA molecule after cation adsorption, $E_{EDTA}$ (eV) and $E_M$ (eV) are the steady state energy of EDTA molecule and cation, respectively.

## MD simulations

Atomistic molecular dynamics (MD) simulations have been performed in the GROMACS[52] (version 2020.6) simulation package. Supplementary Fig. 29 shows our MD simulation model, two $12 \times 12 \times 1$ $Ti_3C_2(OH)_2$ periodic supercell with anions, cations and water molecules. Salt ions were randomly placed in the feed chamber (left side of the sheets) and the whole system was filled with water molecules, while a row of oxygen atoms at the exit of the channel was exposed (otherwise the ion permeation rate would be too low to be detect, Supplementary Fig. 29). Two sheets of surfaces were placed at d-spacing (Supplementary Fig. 12) of 18.4 Å and 16.0 Å for the unintercalated and intercalated by EDTA molecule surfaces respectively. The MLM-EDTA channel models with different charge densities were achieved by controlling the number of hydroxyl and carboxyl groups deprotonations within the channel. The parameters for $Ti_3C_2(OH)_2$ used the Universal force field (UFF), which covers the whole periodic table. The SPC/E model was used for water molecules and the OPLSAA force field was used for ions and EDTA. The interactions between ions and the surface were calculated as the summation of the Lennard-Jones and Electrostatic interactions between all ions and atoms in the surfaces. Before the ion diffusion study, cations and anions were first randomly inserted into the feed chamber (left side of the sheets) and the whole systems were solvated with water molecules, then ran NPT MD simulations for 2 ns to equilibrate the water molecule with ion positions fixed inside the chamber. The resulting system size (cubic box of water molecules) was around $11.0 \times 3.70 \times 3.44$ nm and $11.0 \times 3.70 \times 3.68$ nm, respectively. Specifically, the total number of cations and water molecules in the feed chamber were 20 and 1000, respectively. The number of $Cl^-$ is 20 or 40, which depends on the cations is $K^+$ or $Mg^{2+}$. The ion concentration in feed chamber is defined as the number of ion (in molar) divided by the total volume of the feed chamber. As a result, the ion concentration of the two tested metal ions in the feed chamber was -0.2 M, which is close to the experimental value. Finally, molecule dynamics simulation of 20 ns was performed to investigate the ions passing through the interlayer space of two sheets to the other pure water side. The temperature was controlled by the Nose−Hoover coupling method and a time step of 1 fs was used for the integrations. A cutoff length of 1.2 nm was implemented for the non-bonded interactions, and the Particle Mesh Ewald method with a Fourier spacing of 0.1 nm was applied for the long-range electrostatic interactions. All covalent bonds with hydrogen atoms were constraint using the LINCS algorithm. The snapshots of MD simulation were obtained by VMD software package, version 1.9.3[53].

## Data availability

The data supporting the key findings of this study are available within the article and the Supplementary Information or available from the corresponding authors upon request. All data generated in this study are provided in the Supplementary Information/Source Data file. Source data are provided with this paper.

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

## Acknowledgements

This work was financially supported by the National Key R&D Program of China (2021YFA1201700, W.Z.), National Natural Science Foundation of China (U22A20403, W.Z.), Australian Research Council for funding the Industry Transformation Research Hub for Energy-efficient Separation (IH170100009, X.Z.) and Australian Research Council ARC Future Fellowship (FT210100593, X.Z.). The authors thank Sikai Cheng, Ruolin Lv and Mingyang Li for their assistance in ultrathin section for membrane cross-section TEM images, XPS measurements and DFT calculations, respectively. The authors thank the High-Performance Computing Center (HPCC) of Nanjing University for using the computation resources.

## Author contributions

R.X., W.Z., and X.Z. conceived the idea. W.Z. and X.Z. supervised the study. R.X. designed and conducted the experiments, made and characterized the samples, finished the DFT calculation and MD simulation and wrote the manuscript. W.Z., X.Z., B.P., and Y.K. revised the manuscript. B.P., X.Z., and W.Z. discussed the results and commented on the manuscript.

## Competing interests

The authors declare no competing interests.
