## [Peer Review File · Nature Communications]

Two-dimensional MXene Membranes with Biomimetic Sub-nanochannels for Enhanced Cation SievingREVIEWER COMMENTS

Reviewer #1 (Remarks to the Author):

Membranes with high ion-ion selectivity are highly demanded in many fields. In this work, the authors creatively designed and prepared two-dimensional MXene membranes with ion recognition functionality, based on the structural characteristics of biological potassium ion channel. The structure of the membranes was comprehensively characterized. The authors conducted extensive experimental and computational demonstrations of the mono/bivalent ion selectivity mechanism. The membranes exhibited excellent mono-/bivalent ion selectivity. Interestingly, they observed a correlation between the charge density and ion selectivity. The proposed "entry-passing dominant transport mechanism" provides new insights into ion transport in charged sub-nanochannels. This work should be of great interest for the broad community of ion channels, membrane science, and ions sieving. Therefore, I recommend publication after minor revision:

Below are my specific comments:

λ In Fig. 1f, the full width at the half maximum (FWHM) of the XRD peaks of the MLM and MLM-EDTA membranes are different. In general, the FWHM is related to the stacking mode of nanosheets. A discussion on the influence of EDTA molecules on the stacking of MXene nanosheets is needed.

λ No obvious swelling was observed for MLM-EDTA. More detailed discussion should be included on how EDTA molecule improve the anti-swelling ability of the MLM-EDTA membrane?

λ Fig. 3a shows that the charge density of pure MXene membrane (MLM-EDTA-0) membrane also changes with increasing pH. How to distinguish the contributions of MXene nanosheets and EDTA molecules to the charge density and ion selectivity of MLM-EDTA membrane?

λ As shown in Fig. 4e, increasing charge density leads to more dehydration for cations, which will promote the electrostatic attraction between cations and negatively charged membrane channels. Enhanced electrostatic attraction could promote ion transport as reported in some literature. Why did this study observe opposite phenomena that cation transport in MLM-EDTA membrane was inhibited at higher charge density?

λ Minor remarks: the x subscript in Ti_3C_2Tx is sometimes italic and sometimes not, e.g., P. 3, line 28 and P. 4, line 15.

Reviewer #2 (Remarks to the Author):

The authors have successfully constructed an MLM-EDTA membrane that mimics biological potassium ion channels, achieved ultrahigh K^+ selectivity and satisfactory ion permeation rate. Based on the highly stable two-dimensional channel of MLM-EDTA, the ion transport mechanism through MLM-EDTA under different charge densities was further explored. This manuscript is excellently written, the authors profound understanding and analysis of the

ion transport mechanism are impressive. This study provides novel insights for the design of high-performance ion separation membranes. Therefore, I would like to recommend accepting this manuscript after addressing the following minor issues.

1. Page 6, line 23. The authors should clarify what Mn^+ represents.
2. In Supplementary Fig. 15. Why the stability of MLM-EDTA is better than that of MLM?
3. In Fig. 3e. The authors defined ion transport resistance as the reciprocal of ion permeation rates, the corresponding ion permeation rates data should provide in Supplementary file.
4. In Fig. 3 and Fig. 4d. The conclusion drawn from the authors experiments and simulations is that an increase in charge density inhibits ion transport. But, as far as I know, some research results (e.g., *Sci. Adv.* 8, eabm9436 (2022). *J. Membr. Sci.* 459, 169–176 (2014)) indicate that stronger affinity is beneficial for ions to enter the pore channel, facilitating transport, and therefore promoting ion transport. The authors should provide a detailed explanation.
5. In Fig. 3g. The schematic diagrams of EDTA molecules with different proton numbers in Fig. 3g are not clearly visible. It is suggested to highlight the hydrogen atoms so that easier for readers to understand.
6. Minor corrections. The unit of concentration, Page 4, line 16 and Page 13, line 20.

Reviewer #3 (Remarks to the Author):

The manuscript presents the design, fabrication, and characterization of a bioinspired MXene-based membrane with sub-nanometer pores. Specifically, cation-sieving membranes were constructed using MXene nanosheets and Ethylenediaminetetraacetic acid (EDTA) as building blocks, and the membranes showed mono/divalent cation (K^+/Mg^{2+}) selectivity as high as 121.2. Additionally, molecular dynamics simulations revealed the cation recognition effect of EDTA and partial dehydration effects play critical roles in enhancing the membrane cation selectivity. Overall, the current study provides a new strategy for designing bioinspired membranes with high ion selectivity. The manuscript was well-written, and the contents were well-organized and presented. However, some issues need to be addressed before the manuscript can be accepted for publication. Below are my specific comments:

1. The novelty of the study needs to be more clearly emphasized in the Introduction and during the discussion, to distinguish the current study from the existing studies in the literature.
2. Page 5, Line 23-26: “This limited selectivity could be attributed to the relative large channel size (8.4 Å, Fig. 1g and Supplementary Fig. 13) that barely distinguished between ions via size exclusion (8.24 Å for Ca^{2+} , 8.56 Å for Mg^{2+} , Supplementary Table 1), or the shrinkage of the layers caused by divalent cations intercalated results in slower permeation rates for divalent cations⁴¹.” The authors claimed that the shrinkage of the layers caused by divalent cation can affect the divalent cation permeation rate. However, no shrinking of the MXene layers could be observed when divalent cations were added (according to Figure 1g).
2. When calculating the charge density of the different membranes, were the charges

provided by EDTA included? The charges from EDTA can be considered inside the membrane nanochannels, however, they are different from the charges on the surface of the MXene nanosheets,

3. "The feed side contained 0.2 M five mixed salts (KCl, NaCl, LiCl, CaCl₂, MgCl₂)". This description can be delusive. Please be specific about the concentration of each salt or ion.

4. The thickness of single-layered MXene nanosheets was measured to be 1.5 nm by AFM (Supplementary Fig.2a). However, during the calculation of the membrane free spacing, a thickness of 1 nm was assumed (Supplementary Fig. 12 and Fig. 1g).

5. Supplementary Fig.8: Why was the element N detected in the PES support by the EDX?

6. There are errors in the manuscript that need to be fixed. For example: "the relative large channel" should be "the relatively large channel" (Page 5); "to obtained channel size" should be "to obtain channel size" (Page 2).

Reviewer #4 (Remarks to the Author):

In this manuscript, the authors propose a host-guest strategy using MXene nanosheets and Ethylenediaminetetraacetic acid (EDTA) molecules as building blocks to construct biomimetic sub-nanochannel membrane (MLM-EDTA) with similar channel size and binding sites to those of biological KcsA K⁺ channels. The resulting MLM-EDTA membrane achieved ultrahigh K⁺/Li⁺ and K⁺/Mg²⁺ selectivity using mixed cation solutions and transcended the current upper limit. The manuscript was well written and organized. The authors provided a comprehensive characterization on the physicochemical properties of the membrane, and thoroughly discussed the ion transport and separation mechanisms, supported by solid experimental and simulation results. This interesting work is expected to stimulate novel design of ion separation membranes and advance the understanding of mass transport within sub-nanometer confinement channels. Additionally, the authors provided extensive description on experimental details, such as Ti₃C₂T_x nanosheets preparation, membranes fabrication and membrane encapsulation, which is of great significance for researchers who intend or begin to engage in membrane science studies. Overall, I would like to recommend its acceptance by Nature Communication after a minor revision to address the following specific comments.

1. Many two-dimensional materials have been explored for membrane fabrication. It is better to explain the advantages of using MXene in the introduction.
2. As shown in Fig. 2b, the permeation rates of the monovalent ions crossing the MLM-EDTA membrane are quite different. Detailed explanation should be provided.
3. Separation of monovalent ions is of great significance. I am curious about the impact of charge density on monovalent ions separation.
4. Does charge density influence the effective channel size?
5. How strong is the binding between EDTA and MXene nanosheet? Will EDTA be washed out?

Response Letter

Title: Two-dimensional MXene Membranes with Biomimetic Sub-nanochannels for Enhanced Cation Sieving

Manuscript ID: NCOMMS-23-12011

We greatly appreciate the valuable comments from reviewers, and have conducted additional experiments and explanations to address these questions. The reviewers' comments and questions are responded point to point in this letter.

Reviewer #1 (Remarks to the Author):

Membranes with high ion-ion selectivity are highly demanded in many fields. In this work, the authors creatively designed and prepared two-dimensional MXene membranes with ion recognition functionality, based on the structural characteristics of biological potassium ion channel. The structure of the membranes was comprehensively characterized. The authors conducted extensive experimental and computational demonstrations of the mono/bivalent ion selectivity mechanism. The membranes exhibited excellent mono-/bivalent ion selectivity. Interestingly, they observed a correlation between the charge density and ion selectivity. The proposed "entry-passing dominant transport mechanism" provides new insights into ion transport in charged sub-nanochannels. This work should be of great interest for the broad community of ion channels, membrane science, and ions sieving. Therefore, I recommend publication after minor revision:

Authors: We are greatly thankful for the reviewer's highly positive remarks on our manuscript.

Q1. In Fig. 1f, the full width at the half maximum (FWHM) of the XRD peaks of the MLM and MLM-ETDA membranes are different. In general, the FWHM is related to the stacking mode of nanosheets. A discussion on the influence of ETDA molecules on the stacking of MXene nanosheets is needed.

Authors: We thank the reviewer for suggestions. We have added the following discussion into

the revised manuscript.

Page 5, Line 12-15. “It is noteworthy that the full width at the half maximum (FWHM) of the XRD peaks for membranes in saline solution decreased dramatically after modified with EDTA, indicating the $Ti_3C_2T_x$ nanosheets become more compact and aligned due to the interactions between EDTA and nanosheets^{43, 44}.”

Q2. No obvious swelling was observed for MLM-EDTA. More detailed discussion should be included on how EDTA molecule improve the anti-swelling ability of the MLM-EDTA membrane?

Authors: We thank the reviewer for suggestion. We have added the following discussions on how EDTA molecules improve the anti-swelling properties of MLM-EDTA membranes.

Page 5, Line 17-19. “The significant improvement in the anti-swelling property of MLM-EDTA in saline solution is attributed to the extensive covalent and hydrogen bonding interactions between EDTA molecules and $Ti_3C_2T_x$ nanosheets, which bridge neighboring nanosheets together.”

Q3. Fig. 3a shows that the charge density of pure MXene membrane (MLM-EDTA-0) membrane also changes with increasing pH. How to distinguish the contributions of MXene nanosheets and EDTA molecules to the charge density and ion selectivity of MLM-EDTA membrane?

Authors: We thank the reviewer for question. We agree with the reviewer that the charge density change of MLM-EDTA is stemmed from the deprotonation degree of both $Ti_3C_2T_x$ -affiliated hydroxyl and the anchored EDTA. As shown in Fig. 3c, although the K^+/Mg^{2+} selectivity of both MLM and MLM-EDTA increased when changing the charge density via increasing pH, the selectivity of MLM-EDTA was almost 10 times higher than that of MLM at all pH values. Therefore, we can conclude that the ion selectivity enhancement was mainly contributed by EDTA.

Fig.3c pH dependent K⁺/Mg²⁺ selectivity of MLM and MLM-EDTA-1.5.

Q4. As shown in Fig. 4e, increasing charge density leads to more dehydration for cations, which will promote the electrostatic attraction between cations and negatively charged membrane channels. Enhanced electrostatic attraction could promote ion transport as reported in some literature. Why did this study observe opposite phenomena that cation transport in MLM-EDTA membrane was inhibited at higher charge density?

Authors: We thank the reviewer for question. This could be caused by a number of reasons. Firstly, the distinct experimental phenomena can be attributed to the differences in experimental conditions and membranes properties. A few previous studies also reported instances where an increase in charge density inhibited ion transport (Nat. Mater. 2020, 19, 767-774; ACS Nano 2021, 15, 1240–1249; J. Am. Chem. Soc. 2020, 142, 9827–9833). Secondly, based molecular dynamics simulations (Fig. 4d), although the increase of charge density results in a reduction of the time required for ions to enter the channel, yet on the other hand, it significantly prolongs the duration of ion residence within the channel, thereby inhibiting ion transport. In principle, the impact of increased local charge density within sub-nanochannel on ion transport rates depends on two factors, (1) whether the rate-limiting step for ion transport is the entry of ions into the channel or the diffusion in the channel; (2) whether the driving force for ion transport exceeds the affinity between the charge-providing groups and the ions so that ions can be easily desorbed.

Fig. 4d Average entry time and passing time of K⁺ and Mg²⁺ passing MLM-EDTA systems with different number of negatively charged oxygen atoms.

Q5. Minor remarks: the x subscript in Ti₃C₂T_x is sometimes italic and sometimes not, e.g., P. 3, line 28 and P. 4, line 15.

Authors: We thank the reviewer for the guidance and apologize for the typo. We have double checked the full manuscript and made corrections.

Reviewer #2 (Remarks to the Author):

The authors have successfully constructed an MLM-EDTA membrane that mimics biological potassium ion channels, achieved ultrahigh K⁺ selectivity and satisfactory ion permeation rate. Based on the highly stable two-dimensional channel of MLM-EDTA, the ion transport mechanism through MLM-EDTA under different charge densities was further explored. This manuscript is excellently written, the authors profound understanding and analysis of the ion transport mechanism are impressive. This study provides novel insights for the design of high-performance ion separation membranes. Therefore, I would like to recommend accepting this manuscript after addressing the following minor issues.

Authors: We are greatly thankful for the reviewer's highly positive remarks on our manuscript.

Q1. Page 6, line 23. The authors should clarify what Mⁿ⁺ represents.

Authors: We thank the reviewer for the suggestion, Mⁿ⁺ in manuscript has been described.

Q2. In Supplementary Fig. 15. Why the stability of MLM-EDTA is better than that of MLM?

Authors: We thank the reviewer for question. The eight carboxylic oxygen atoms of EDTA molecules readily form numerous hydrogen bonds with hydrogen atoms on the benzene ring of PES substrate, while there are also abundant covalent and hydrogen bonding interactions between EDTA molecules and MXene nanosheets. Consequently, EDTA functions as an adhesive to firmly attach the nanosheets onto PES substrate and link neighboring nanosheets, thereby improving the stability of MLM-EDTA over that of MLM.

Q3. In Fig. 3e. The authors defined ion transport resistance as the reciprocal of ion permeation rates, the corresponding ion permeation rates data should provide in Supplementary file.

Authors: We thank the reviewer for suggestion, the corresponding ion permeation rates data has been added to the Supplementary file as Supplementary Fig. 23.

Supplementary Fig. 23 K⁺ and Mg²⁺ permeation rates through MLM-EDTA-1.5 with different charge densities.

Q4. In Fig. 3 and Fig. 4d. The conclusion drawn from the authors experiments and simulations is that an increase in charge density inhibits ion transport. But, as far as I know, some research results (e.g., Sci. Adv. 8, eabm9436 (2022). J. Membr. Sci. 459, 169–176 (2014)) indicate that stronger affinity is beneficial for ions to enter the pore channel, facilitating transport, and therefore promoting ion transport. The authors should provide a detailed explanation.

Authors: We thank the reviewer for question. We have read the two studies mentioned by the reviewers in detail, and we believe that our findings are not contradictory to the two studies.

Firstly, in the first study (Sci. Adv. 8, eabm9436 (2022)), the experimental results shows that ion flux decreases as pH increases (Fig. S9 from reference [1]), which is consistent with our conclusion that an increase in charge density inhibits ion transport. Secondly, researchers proposed three limitations for the selective transportation of ions with stronger affinity, namely, (1) strong driving force is required for the repeated adsorption and desorption of stronger affinity ions (original sentence “While we expect that the conceptual findings translate to other membrane chemistries, there are some notable exceptions to when these methods may be effectively applied. For example, membranes that irreversibly bind target species are unlikely to exhibit selectivity for that species, perhaps unless an external driving force is applied to overcome the desorption energy barrier.”), (2) it is difficult to improve ions selectivity when the rate-determining transport step for weaker affinity ions is intramembrane diffusion (original sentence “Moreover, membranes with low sorption selectivity, which are specifically unable to substantially hinder a weaker binding species from entering the membrane, will not have improved selectivity with smaller thicknesses when the rate-determining transport step of a weaker binding species is diffusion through the membrane.”), (3) the membrane which achieves ion selective separation solely based on the difference in affinity is not suitable for the mono-/divalent ions separation (original sentence “A limitation, however, is that a pre- or postseparation step would be necessary to isolate divalent cations from monovalent cations, as membranes with coordination chemistry are unlikely to adequately reject small, monovalent species while removing divalent species.”). As for the second study (J. Membr. Sci. 459, 169–176 (2014)), its system and conclusion are analogous to those of the first research.

In our study, sub-nanometer channels significantly enhance the affinity between ions and the channel, and there is strong affinity between EDTA molecules and divalent cations. Consequently, rapid adsorption and desorption of divalent ions within the channel without additional driving forces is difficult to achieve. In addition, simulation results suggest that diffusion inside channels is the rate-limiting step for ion transport through MLM-EDTA. Meanwhile, previous studies reported that the higher binding affinity between ion and channel results in greater resistance to ion transport (Angew. Chem. Int. Ed. 2021, 60, 22265-22269; Angew. Chem. Int. Ed. 2016, 55, 15120-15124).

Reference

[1] DuChanois, R. M. et al. Designing polymeric membranes with coordination chemistry for high-precision ion separations. *Sci. Adv.* 8, eabm9436 (2022).

Q5. In Fig. 3g. The schematic diagrams of EDTA molecules with different proton numbers in Fig. 3g are not clearly visible. It is suggested to highlight the hydrogen atoms so that easier for readers to understand.

Authors: We thank the reviewer for suggestion, Fig. 3g has been changed to the following figure.

Fig. 3g The DFT calculated adsorption energy for K⁺ and Mg²⁺ on EDTA molecules with different negatively charged oxygen atoms.

Q6. Minor corrections. The unit of concentration, Page 4, line 16 and Page 13, line 20.

Authors: We apologize for the typo, all concentration units in the manuscript have been unified and corrected.

Reviewer #3 (Remarks to the Author):

The manuscript presents the design, fabrication, and characterization of a bioinspired MXene-based membrane with sub-nanometer pores. Specifically, cation-sieving membranes were constructed using MXene nanosheets and Ethylenediaminetetraacetic acid (EDTA) as building blocks, and the membranes showed mono/divalent cation (K⁺/Mg²⁺) selectivity as high as 121.2. Additionally, molecular dynamics simulations revealed the cation recognition effect of

EDTA and partial dehydration effects play critical roles in enhancing the membrane cation selectivity. Overall, the current study provides a new strategy for designing bioinspired membranes with high ion selectivity. The manuscript was well-written, and the contents were well-organized and presented. However, some issues need to be addressed before the manuscript can be accepted for publication. Below are my specific comments:

Authors: We are greatly thankful for the reviewer's highly positive remarks on our manuscript.

Q1. The novelty of the study needs to be more clearly emphasized in the Introduction and during the discussion, to distinguish the current study from the existing studies in the literature.

Authors: We thank the reviewer for suggestion. The novelty of the study was further emphasized in the Introduction and Discussion as follows:

In Introduction,

Page 2, Line 13-17. "While these designs captured one of biological ion channel feature, to replicate all the three in high precision remain grandly challenging. Moreover, the transport and separation mechanisms of ions within sub-nanochannels modulated by local charge density remains elusive and an effective strategy to fabricate high-efficiency mono-/divalent ions selective membranes is yet to be developed."

In Discussion,

Page 7, Line 4-6. "Notably, MLM-EDTA achieving excellent K^+/Mg^{2+} selectivity of 121.2 using mixed salt solution as the feed, which outperforms reported previously membranes under similar measuring conditions and transcends the current upper limit (Fig. 2f and Supplementary Table 2)."

Page 11, Line 21-24. "Meanwhile, an entry-passing dominant transport mechanism in 2D sub-nanochannels is proposed, the cations selectivity is dominated by the entry effects in a low charge density channel, whereas in the high charge density channel, the cation selectivity is dominated by the passing effects (Fig. 4f)."

Q2. Page 5, Line 23-26: "This limited selectivity could be attributed to the relative large channel size (8.4 Å, Fig. 1g and Supplementary Fig. 13) that barely distinguished between ions via size

exclusion (8.24 Å for Ca²⁺, 8.56 Å for Mg²⁺, Supplementary Table 1), or the shrinkage of the layers caused by divalent cations intercalated results in slower permeation rates for divalent cations⁴¹.” The authors claimed that the shrinkage of the layers caused by divalent cation can affect the divalent cation permeation rate. However, no shrinking of the MXene layers could be observed when divalent cations were added (according to Figure 1g).

Authors: We thank the reviewer for the question. As shown in the upper part of Fig. 1g, after soaking in water, KCl, NaCl, LiCl, CaCl₂, MgCl₂ and mixed salt solutions, the free spacing of MLM is 9.2, 8.4, 7.0, 7.3, 7.1, 7.4, 8.4 Å respectively. The free spacings of MLM in CaCl₂ and MgCl₂ solution are smaller than that in water, indicates the addition of divalent cations shrinks the free spacing of MXene layers. But we agree with the reviewer that this sentence is confusing. So, we have removed this sentence from the revised manuscript.

Fig. 1g The free-spacings of MLM and MLM-EDTA-1.0 in the dry state and various solutions.

Q3. When calculating the charge density of the different membranes, were the charges provided by EDTA included? The charges from EDTA can be considered inside the membrane nanochannels, however, they are different from the charges on the surface of the MXene nanosheets,

Authors: We thank the reviewer for the question. The charges provided by EDTA molecules were included in the charge density calculation. We calculated the charge density from the zeta potential of the membrane (Equation (2)), and the result is shown in Fig. 3a. The gap between MLM and MLM-EDTA-1.5 is the charges provided by the EDTA molecules. In the manuscript we emphasized that the charge density inside of MLM-EDTA nanochannels mainly comes from EDTA molecules and the surface of MXene nanosheets (Fig. 3b). We agree with the reviewer

that the charges from EDTA molecules are different with that from the surface of the MXene nanosheets. Therefore, we conducted a blank control experiment at different pH values to distinguish the contributions of MXene nanosheets and EDTA molecules to the ion selectivity of MLM-EDTA membrane. As shown in Fig. 3c, although the K^+/Mg^{2+} selectivity of both MLM and MLM-EDTA increased when changing the charge density via increasing pH, but the selectivity of MLM-EDTA was almost 10 times higher than that of MLM at all pH values. Therefore, we can conclude that the ion selectivity enhancement was mainly contributed by EDTA.

Fig. 3 **a** Charge densities of MLM and MLM-EDTA-1.5 membranes at different pH, calculated by the membranes surface zeta potential (equation (2)). **b** Schematic diagram of the main sources of negative charges in MLM-EDTA. **c** pH dependent K^+/Mg^{2+} selectivity of MLM and MLM-EDTA-1.5.

The membrane charge density (σ , $C \cdot m^{-2}$) can be calculated using an Gouy–Chapman equation:

$$\sigma = -\varepsilon\kappa\zeta \frac{\sinh\left(\frac{F\zeta}{2RT}\right)}{\frac{F\zeta}{2RT}} \quad (2)$$

$$\kappa^{-1} = \left(\frac{\varepsilon RT}{2F^2 C_0}\right)^{1/2} \quad (3)$$

where ε ($6.933 \times 10^{-10} F \cdot m^{-1}$) is permittivity, κ^{-1} (m) is Debye length, ζ (V) is membrane zeta potential, F ($96485 C \cdot mol^{-1}$) is Faraday constant, R ($8.3145 J \cdot mol^{-1} \cdot K^{-1}$) is gas constant, C_0 ($mol \cdot m^{-3}$) is the concentration of the electrolyte.

Q4. “The feed side contained 0.2 M five mixed salts (KCl, NaCl, LiCl, CaCl₂, MgCl₂)”. This description can be delusive. Please be specific about the concentration of each salt or ion.

Authors: We apologize for the confusing description, and the expression has been corrected to

the following sentence:

“The feed side contained five mixed salts (KCl, NaCl, LiCl, CaCl₂, MgCl₂), each salt concentration is 0.2 M”

Q5. The thickness of single-layered MXene nanosheets was measured to be 1.5 nm by AFM (Supplementary Fig.2a). However, during the calculation of the membrane free spacing, a thickness of 1 nm was assumed (Supplementary Fig. 12 and Fig. 1g).

Authors: We thank the reviewer for questions. The thickness of single-layered MXene nanosheets measured by AFM is slightly larger than theoretically predicted value of 1.0 nm, which can be attributed to water molecules adsorbed on the surface [2, 3]. To obtain the true thickness of the MXene nanosheets, we added STEM measurements on the MLM-EDTA membrane cross-section, the results confirms that the nanosheets thickness is about 1 nm. The new results have been added to the supplementary information in Supplementary Fig. 11c.

Supplementary Fig. 11 (c) STEM image of the cross-section of the MLM-EDTA-1.0.

Reference

[2] Ding, L. et al. MXene molecular sieving membranes for highly efficient gas separation. *Nat. Commun.* 9, 155 (2018).

[3] Lipatov, A. et al. Effect of Synthesis on Quality, Electronic Properties and Environmental Stability of Individual Monolayer Ti₃C₂ MXene Flakes. *Adv. Electron. Mater.* 2, 1600255 (2016).

Q6. Supplementary Fig.8: Why was the element N detected in the PES support by the EDX?

Authors: We thank the reviewer for question. The MLM-EDTA membrane was obtained by suction filtering a dispersion solution containing EDTA molecules and MXene nanosheets, and a small amount of EDTA molecules attaching to the pores of the PES support during the suction filtration process, so a weak N element signal was detected on the PES. We have added ion separation measurements on pure PES support and PES-EDTA support. As shown in Fig. R1, the pure PES support and PES-EDTA do not have the capability of ion separation.

Fig. R1 K^+/Mg^{2+} selectivity of PES, PES-EDTA and MLM-EDTA.

Q7. There are errors in the manuscript that need to be fixed. For example: “the relative large channel” should be “the relatively large channel” (Page 5); “to obtained channel size” should be “to obtain channel size” (Page 2).

Authors: We thank the reviewer for the guidance. We have double checked the full manuscript and corrected typos and errors.

Reviewer #4 (Remarks to the Author):

In this manuscript, the authors propose a host-guest strategy using MXene nanosheets and Ethylenediaminetetraacetic acid (EDTA) molecules as building blocks to construct biomimetic sub-nanochannel membrane (MLM-EDTA) with similar channel size and binding sites to those of biological KcsA K^+ channels. The resulting MLM-EDTA membrane achieved ultrahigh K^+/Li^+ and K^+/Mg^{2+} selectivity using mixed cation solutions and transcended the current

upper limit. The manuscript was well written and organized. The authors provided a comprehensive characterization on the physicochemical properties of the membrane, and thoroughly discussed the ion transport and separation mechanisms, supported by solid experimental and simulation results. This interesting work is expected to stimulate novel design of ion separation membranes and advance the understanding of mass transport within sub-nanometer confinement channels. Additionally, the authors provided extensive description on experimental details, such as Ti₃C₂T_x nanosheets preparation, membranes fabrication and membrane encapsulation, which is of great significance for researchers who intend or begin to engage in membrane science studies. Overall, I would like to recommend its acceptance by Nature Communication after a minor revision to address the following specific comments.

Authors: We are greatly thankful for the reviewer's highly positive remarks on our manuscript.

Q1. Many two-dimensional materials have been explored for membrane fabrication. It is better to explain the advantages of using MXene in the introduction.

Authors: We thank the reviewer for suggestion. We have added the following statements regarding the advantages of MXene in the introduction.

Page 2, Line 26-28. "MXene membranes have shown excellent thermal stability, mechanical flexibility, antibacterial properties, and the ability for scalable manufacturing, indicating their potential for applications in water treatment^{25, 27, 38, 39}."

Q2. As shown in Fig. 2b, the permeation rates of the monovalent ions crossing the MLM-EDTA membrane are quite different. Detailed explanation should be provided.

Authors: We thank the reviewer for suggestion. We have added the following explanation to the Discussion.

Page 7, Line 1-4. "Meanwhile, the differential affinity of EDTA molecules towards monovalent cations endows MLM-EDTA with superior capability for separating monovalent ions, achieving a K⁺/Li⁺ selectivity up to 5.4 (Supplementary Fig. 14a). This suggests that rational design of affinity groups inside sub-nanochannels may lead to highly selective separation of monovalent ions."

Q3. Separation of monovalent ions is of great significance. I am curious about the impact of charge density on monovalent ions separation.

Authors: We thank the reviewer for the question. As shown in Supplementary Fig. 25, the charge densities and the K^+/Li^+ selectivity are linearly correlated. The increasing charge density led to pronounced increase in Li^+ transport resistance and a negligible rise in K^+ transport resistance, thus enhancing K^+/Li^+ selectivity. This is similar to the K^+/Mg^{2+} separation results, but with a much smaller effect of charge density on selectivity. This may be due to the smaller difference in affinity between EDTA molecules and monovalent cations. The new data has been added to the supplementary information.

Supplementary Fig. 25 Relationship between K^+ , Li^+ transport resistance (Reciprocal of ion permeation rates, $1/P_i$) and K^+/Li^+ selectivity with charge density as independent variable for MLM-EDTA-1.0.

Q4. Does charge density influence the effective channel size?

Authors: We thank the reviewer for question. We believe that charge density has a minimal impact on the effective channel size, which can be attributed to two reasons. First, the interlayer spacing of the MLM-EDTA remained unchanged under different pH conditions (Supplementary Fig. 20). Second, tuning of charge density was achieved by controlling the different

deprotonation levels of hydroxyl groups on MXene surfaces and EDTA carboxylic groups. However, the number of hydroxyl and carboxylic groups are limited and difficult to affect the entire nanochannel, only minor changes in local positions of the channel are expected.

Supplementary Fig. 20 XRD patterns of MLM-EDTA-1.5 in different pH values mixed salts (KCl, NaCl, LiCl, CaCl₂ and MgCl₂) solutions.

Q5. How strong is the binding between EDTA and Mxene nanosheet? Will EDTA be washed out?

Authors: We thank the reviewer for questions. The interaction between EDTA and MXene nanosheets is quite strong, which can be attributed to two reasons. First, characterization techniques such as XPS and FTIR demonstrate the existence of a large number of covalent bonds and hydrogen bonding interactions between EDTA molecules and MXene nanosheets. Second, the stable interlayer structure of the MLM-EDTA membrane in different salt solutions (Fig. 1g) suggests that EDTA molecules can anchor within the MXene nanosheets interlayer. Moreover, the strong binding is further confirmed by the excellent cycling stability shown in supplementary Figure 16.

Fig. 1g The free-spacings of MLM and MLM-EDTA-1.0 in the dry state and various solutions.

Supplementary Fig. 16 Cycling performance of MLM-EDTA-1.0.

REVIEWERS' COMMENTS

Reviewer #1 (Remarks to the Author):

Membranes with high ion-ion selectivity are highly demanded in many fields. In this work, the authors creatively designed and prepared two-dimensional MXene membranes with ion recognition functionality, based on the structural characteristics of biological potassium ion channel. I do believe that the work is very important for the ions separation field.

Reviewer #2 (Remarks to the Author):

The authors have addressed all comments by this reviewer. I would like to recommend acceptance of this paper in its current form.

Reviewer #3 (Remarks to the Author):

All my comments have been addressed by the authors, and I do not have any additional comments. The manuscript in its current form can be accepted for publication.

Response Letter

Title: Two-dimensional MXene Membranes with Biomimetic Sub-nanochannels for Enhanced Cation Sieving

Manuscript ID: NCOMMS-23-12011A

Reviewer #1 (Remarks to the Author):

Membranes with high ion-ion selectivity are highly demanded in many fields. In this work, the authors creatively designed and prepared two-dimensional MXene membranes with ion recognition functionality, based on the structural characteristics of biological potassium ion channel. I do believe that the work is very important for the ions separation field.

Authors: We thank the reviewer for the positive recommendation.

Reviewer #2 (Remarks to the Author):

The authors have addressed all comments by this reviewer. I would like to recommend acceptance of this paper in its current form.

Authors: We thank the reviewer for the positive recommendation.

Reviewer #3 (Remarks to the Author):

All my comments have been addressed by the authors, and I do not have any additional comments. The manuscript in its current form can be accepted for publication.

Authors: We thank the reviewer for the positive recommendation.